# Research on the Behavior of Stiffening Walls in Single-Storey Buildings Made of Autoclaved Aerated Concrete (AAC) Masonry Units

**DOI:** 10.3390/ma15207404

**Published:** 2022-10-21

**Authors:** Krzysztof Grzyb, Radosław Jasiński

**Affiliations:** Department of Building Structures, Faculty of Civil Engineering, Silesian University of Technology, Akademicka 5, 44-100 Gliwice, Poland

**Keywords:** stiffening wall, shear wall, wall stiffness, spatial stiffness, crack morphology, horizontal load, building analysis, masonry structure, autoclaved aerated concrete, digital image correlation

## Abstract

Experimental identification of stiffening walls is often limited to studying single-wall models. However, these samples do not reflect many additional effects—torsion of the building and redistribution of internal forces. This paper presents the results of two full-scale buildings made of autoclaved aerated concrete (AAC) masonry elements. The primary purpose of the work was to determine the changes in the stiffness of the shear walls and to attempt the empirical distribution of loads on the stiffening walls. The intermediate goals were: a description of the crack morphology and the mechanism of failure, the designation of the stiffening walls’ behavior. It was shown that the first crack formed in the tensile corner of the door opening, and the subsequent cracks formed in the wall without a hole. Based on the changes in the value of the shear deformation angles, the phases of work of the stiffening walls were determined. The presented research results are only a part of an extensive study of stiffening walls in masonry buildings conducted at the Silesian University of Technology.

## 1. Introduction

### 1.1. Terminology and Standard Inaccuracies

Information on the design of shear walls in the Eurocode is limited to general recommendations and regulations. According to the definition [1]—term 1.5.10.10, the stiffening wall is a structural element located perpendicular to other walls, which supports them with the transfer of horizontal forces, preventing buckling and contributing to the stability of the building. The stiffening elements mainly take the horizontal loads acting in the plane of the walls. Although these elements are primarily shear in the plane, they can simultaneously transfer other forces (vertical loads, bending moments) depending on other functions and load distribution in the building.

In literature, the concept of stiffening walls sometimes replaces the term shear walls [2,3,4,5]. In the code regulations of Eurocode 6, the definition of a shear wall [1]—term 1.5.10.9 is distinguished and denotes shear walls whose task is to transfer lateral forces acting in the plane of the wall. This terminological inaccuracy raises problems in naming the walls of the building.

The structure of the entire building determines the distribution of loads on the stiffening walls. Horizontal forces can be transferred to walls in proportion to their stiffness [6], provided that the slab act as a rigid diaphragm. Suppose the horizontal forces act in the eccentricity concerning the center of gravity of the stiffening walls, or the layout of the stiffening walls is asymmetrical. In that case, the torsional effect should be considered in the calculations. It is worth mentioning that this situation occurs in practice in most cases. The standard recommendations do not indicate a consistent design method to consider or omit the torsion effect [6]. Moreover, the standard [1] allows for the redistribution of internal forces to a maximum level of 15%—but there is no experimental recognition of the issue.

### 1.2. Stiffening and Shear Walls Test Results

Stiffening walls can be erected in different ways, for instance, as confined walls [7] by making circumferential reinforced concrete surrounding the masonry [8,9,10] or by filling the external masonry elements with mortar (reinforced grouted masonry) [11,12,13,14,15,16]. Moreover, stiffening masonry can be unreinforced [17] or reinforced [18,19]. Some ways of increasing shear rigidity were collected in [20].

Both flexural and shear deformations affect the total horizontal displacement of a single shear wall [21,22]. Stiffening walls with mating sections of transverse walls (flanged walls [23,24]) can take part in taking horizontal loads. The work [25] shows the influence of the flanged parts on the transmission of normal stresses and a slight influence on the transmission of shear stresses caused by the horizontal load.

The level of compressive stress (prestress) influences the behavior of masonry [26]. When the compressive stresses of the wall are low, cracks run along the diagonal of the wall in the head and bed joints of the masonry [27]. The damages open along the diagonal with large horizontal (e.g., 30 mm) displacements [28]. At high compressive stresses of the shear wall, cracks run through the masonry units and have a vertical course locally. If the damage has a stepped path, the reduction in strength caused by tension is smaller than in the case of cracks in the masonry elements [29]—Figure 1. In some cases, there may be a mixed mechanism that combines step failure and cracks running through masonry units [30]. The slenderness of the wall also influences the crack morphology [31].

The complexity of the calculation depends on the localization of the perforation in the building [32]. The geometry of both door and window openings is also essential [33,34]. If the openings are in-line—it is possible to distinguish the stiffening units relatively unequivocally. In the case of the openings’ irregular location, the analysis’s complexity increases significantly. Various analytical models based on flexural or shear failure are used to determine the load capacity of in-plane shear walls [35] (sometimes called SLC—shear load capacity [36]). A simple equilibrium model can be used to estimate the behavior of shear walls [37]. Work [38] proposes mathematical models to predict the in-plane horizontal load capacity of unreinforced masonry (URM) walls, considering possible failure modes and the presence of perforations. Paper [39] presents analytical models for partially grouted masonry.

The analysis depends on the relative stiffnesses of the vertical and horizontal wall sections (spandrels, lintels) between the openings [40] and the building geometry. Usually, the analysis assumes that the walls are fixed in the foundation, but sometimes analysis includes soil-structure interaction [41]. In the case of rigid vertical inter-opening pillars (cantilever piers)—the lintels (spandrels, lintels) act as horizontal struts between the vertical pillars. In this scheme, the determination of internal forces is relatively simple and based on the cantilevered, bent work of the vertical wall fragments between the openings. The role of lintels is to transfer forces to vertical elements, and their deformations correspond and coincide with the rigid vertical parts of the wall.

In the second situation, the vertical wall fragments are characterized by much lower stiffness than the horizontal elements. It is assumed that the lintels and horizontal fragments are infinitely rigid, and the vertical pillars are analyzed as fixed at the base of the columns. The horizontal stripes between the windows are designed for the forces transmitted by vertical elements. The verification of the theory with reality is often based on a single wall test, but these models do not reflect many additional effects.

The experimental spatial model of the building allows taking into account the cooperation between the walls in transferring horizontal loads and capturing the torsion of the building [42,43,44]. A careful analysis of the crack morphology provides information on the actual heights of inter-hole pillars and the recognition of stress concentration areas. Full-scale building studies have the advantage over single shear walls research, but experimental analyses of real-scale [45] spatial models are rare in the literature. The analyses usually are conducted for scaled-down models [46], or when tests are performed on full-scale models, there are focused on seismic loads [47]. Instead, computational analyzes are performed based on, e.g., statistical characterization of mechanical properties of masonry [48] and stochastic discontinuum analysis [49]. The finite element method [50,51] (FEM), discrete macro-element model (DMEM) [52] or multiple vertical line element method (MVLEM) [53] also is used.

This paper presents the results of full-scale buildings made of autoclaved aerated concrete masonry elements [54]. The digital image correlation (DIC) technology was used to observe the propagation of cracks and damages [55,56,57]. The presented research is the basis for further analysis of the stiffening walls’ behavior in the masonry structure’s spatial model.

The primary purpose of the work was to determine the changes in the stiffness of the stiffening walls and to attempt the empirical distribution of loads on the stiffening walls. The intermediate goals were: description of the crack morphology and the mechanism of failure, the designation of the stiffening walls’ behavior. The publication is only a part of extensive research at the Silesian University of Technology on stiffening walls in masonry buildings.

## 2. Materials and Methods

### 2.1. Research Models

Two models of single-storey buildings were made to analyze the spatial behavior of the stiffening walls. The construction of each model was the same. First, the lower reinforced concrete perimeter beam with a cross-section of 20 × 22 cm was made (width x height). This element is reinforced with four rods Ø12 mm and stirrups with a diameter of Ø6 mm with a spacing of 15 cm. The circumferential beam was fastened using screws—Figure 2a. After 28 days, walls were made of autoclaved aerated concrete masonry elements. The dimensions of a single block were 18 × 59 × 24 cm (width × length × height). The walls are designed in such a way as to minimize the need to cut the blocks—the walls were ten layers of masonry elements high, each 4 m long. In one of the walls of the building was a door opening 100 cm wide and 192 cm high (eight layers of masonry blocks). The opening is covered with a prefabricated system lintel made of AAC. The walls were erected with a thin joint without filling the head joints—Figure 2b.

On the walls, precast L-shaped elements were laid, which were made of lightweight concrete. On the fittings, there was reinforcement consisting of Ø12 mm main rods and Ø 6 mm stirrups at a distance of 15 cm, similarly to the bottom beam. Instead of the traditional monolithic slab, a modern slab structure consisting of precast elements was made. Three prefabricated panels 1200 mm wide each with four prestressed ribs were used. In order to reduce the total weight of the slab, polystyrene was inserted between the ribs. The thickness of the precast slab element was 4 cm and 12 cm at the connection with the rib. Reinforcement hooks with a diameter of Ø10 mm with a spacing of 100 cm were built up on the perimeter, which constituted the upper reinforcement for a partial moment. The top surface reinforcement was placed at the very end in the form of Ø4 mm diameter nets with a mesh size of 15 cm—Figure 2c. After the reinforcement was made, the structure was monolithized with a 5 cm thick concrete overlay—Figure 2d. Figure 3 shows the details of the slab structure.

Both models were made of AAC masonry units on a system mortar class M5 with thin-layer bed joints and unfilled head joints. The compressive strength of the wall was determined according to PN-EN 1052-1 [58] and was *f*_c_ = 2.97 N/mm^2^, and the modulus of elasticity was equal to *E*_m_ = 2040 N/mm^2^. The initial shear strength was determined according to the procedures specified in PN-EN 1052-3: 2004 [59] and was *f*_vo_ = 0.31 N/mm^2^. The average coefficient of friction of the mortar in the bed joints was equal to *μ* = 0.92. The shear modulus was determined according to ASTM E519-81 [60] and was equal to G = 329 N/mm^2^.

### 2.2. Test Stand

Stiffening walls are horizontally loaded due to the action of wind, uneven building settlement or paraseismic loads. In order to analyze the behavior of such structures, a dedicated test stand was built—Figure 4.

The research models were erected in the laboratory hall. Next to the building, a steel column with a brace was placed halfway along the length of the walls. The horizontal shear force was induced by a hydraulic actuator supported by steel construction mounted on the steel column. The operating range of the hydraulic jack was 1000 kN. The load was applied to the building at the axis level of the slab. The dynamometer of range 250 kN enabled the measurement of horizontal force.

Moreover, the actual stiffening walls are loaded with dead loads (self-weight and weight of finishing layers) and live loads (resulting from the utility function of the building). It means that during wall shearing, initial compressive stresses occur in it. For inducing initial stress in walls, weights with a diameter of 60 cm and a height of 30 cm were used. The vertical load was suspended on the twelve steel rods, and on each rod hung three weights—Table 1. The view of loads of the models is shown in Figure 5, and the values of initial compressive stresses are in Table 2. The research model in the test stand is presented in Figure 6.

### 2.3. Measurement Methodology

There were mounted measuring bases on each model’s walls (measuring frame). The size of the frame system was designed to cover the largest possible area of the wall and avoid edge disturbances. The frames were rectangular, 3260 mm long, and 2150 mm high—Figure 7. Linear variable differential transformer (LVDT) sensors [61] measured the change in vertical and horizontal segments and the length of diagonals of the measurement base. LVDT sensors with a measuring range of 20 mm (PJX-20) were mounted on the diagonals, LVDT sensors with a range of 10 mm (PJX-10) on the vertical and horizontal frames. The accuracy of the indications was 0.002 mm. The measuring base was attached to the masonry wall point-by-point in the corners with screws. This solution allows the measurement of the shear strain in the elastic range and the deformation angle in the nonlinear range.

Basic lengths of the measuring frame fragments changed by a value of ∆_a_, ∆_b_, ∆_c_, ∆_d_, ∆_e_, ∆_f_. After deformation, the total length of vertical fragments was calculated from Equations (1) and (2); horizontal fragments from Equations (3) and (4) and diagonals from (5) and (6).
(1)$$b_{d}=b_{0}+{∆}_{b}$$
(2)$$d_{d}=d_{0}+{∆}_{d}$$
(3)$$a_{d}=a_{0}+{∆}_{a}$$
(4)$$c_{d}=c_{0}+{∆}_{c}$$
(5)$$e_{d}=e_{0}+{∆}_{e}$$
(6)$$f_{d}=f_{0}+{∆}_{f}$$

The value of the change in the length of the measuring frame fragments was used to determine the partial deformation angles *θ**_i_* (where *i* = 1, 2, 3, 4) isolated from the deformed measuring system—Figure 8. The partial values of the global deformation angle were calculated based on the law of cosines (relations (7)–(10):-the triangle formed by lines *a*_d_, *d*_d_ and *f*_d_:
(7)$$f_{d}^{2}=a_{d}^{2}+d_{d}^{2}-2a_{d}d_{d}\;\cdot \;\cos\left(\frac{\pi }{2}+\theta_{1}\right)\to \;\theta_{1}=-\frac{\pi }{2}+arccos\left(\frac{a_{d}^{2}+d_{d}^{2}-f_{d}^{2}}{2a_{d}d_{d}}\right)$$

-the triangle formed by lines *c*_d_, *d*_d_ and *e*_d_:


(8)
$$e_{d}^{2}=c_{d}^{2}+d_{d}^{2}-2c_{d}d_{d}\;\cdot \;\cos\left(\frac{\pi }{2}-\theta_{2}\right)\to \;\theta_{2}=\frac{\pi }{2}-arccos\left(\frac{c_{d}^{2}+d_{d}^{2}-e_{d}^{2}}{2c_{d}d_{d}}\right)$$


-the triangle formed by lines *a*_d_, *b*_d_ and *e*_d_:


(9)
$$e_{d}^{2}=a_{d}^{2}+b_{d}^{2}-2a_{d}b_{d}\;\cdot \;\cos\left(\frac{\pi }{2}-\theta_{3}\right)\to \;\theta_{3}=\frac{\pi }{2}-arccos\left(\frac{a_{d}^{2}+b_{d}^{2}-e_{d}^{2}}{2a_{d}b_{d}}\right)$$


-the triangle formed by lines *b*_d_, *c*_d_ and *f*_d_:


(10)
$$f_{d}^{2}=b_{d}^{2}+c_{d}^{2}-2b_{d}c_{d}\;\cdot \;\cos\left(\frac{\pi }{2}+\theta_{4}\right)\to \;\theta_{4}=-\frac{\pi }{2}+arccos\left(\frac{b_{d}^{2}+c_{d}^{2}-f_{d}^{2}}{2b_{d}c_{d}}\right)$$


The global value of deformation angle *θ* at the following load levels was calculated as the arithmetic mean of the partial values of deformation angles *θ**_i_* (where *i* = 1, 2, 3, 4)—relation (11).
(11)$$\Theta =\frac{1}{n}\overset{n=4}{\underset{i=1}{\sum }}\left|\Theta_{i}\right|$$

The total angle of strain deformation determined from dependence 11 includes both deformations resulting from in-plane bending (Figure 9) as well as shear (Figure 10). As a result of bending, the lengths of the vertical bases change and horizontal lengths remain without changes. Thanks to that, trigonometric relationships (12) and (13) enable calculating the lengths of diagonals caused by bending moments.
(12)$$c_{1\;}=\sqrt{c_{0}^{2}-{\left(\frac{d_{d}-b_{d}}{2}\right)}^{2}}$$

The diagonal lengths resulting from the flexural deformations can be calculated according to the Formula (13).
(13)$$e_{1\;}=f_{1}=\sqrt{c_{1}^{2}+{\left(d_{d}-\left(\frac{d_{d}-b_{d}}{2}\right)\right)}^{2}}\to \;e_{1\;}=f_{1\;}=\sqrt{c_{0}^{2}+d_{d}^{2}-2\;\cdot \;d_{d}\left(\frac{d_{d}-b_{d}}{2}\right)}$$

Subtracting from the total lengths of the diagonals the lengths of diagonals resulting from only in-plane bending, the differences in the length of the diagonals are obtained—relationships (14) and (15).
(14)$$∆f_{1\;}=f_{d}-f_{1}$$
(15)$$∆e_{1\;}=e_{d}-e_{1}$$

The diagonal lengths resulting from the shear deformations can be calculated according to Formulas (16) and (17).
(16)$$f_{s\;}=f_{0}+∆f_{1\;}$$
(17)$$e_{s\;}=e_{0}+∆e_{1\;}$$

Values of partial angle of strain deformation can find it from the law of cosines:-the triangle formed by lines *d*_0_, *f*_s_ and *a*_d_:
(18)$$f_{s}^{2}=d_{0}^{2}+a_{d}^{2}-2d_{0}a_{d}\;\cdot \;\cos\left(\frac{\pi }{2}+\theta_{1s}\right)\to \;\theta_{1s}=-\frac{\pi }{2}+arccos\left(\frac{d_{0}^{2}+a_{d}^{2}-f_{s}^{2}}{2d_{0}a_{d}}\right)$$

-the triangle formed by lines *c*_d_, *e*_s_ and *d*_0_:


(19)
$$e_{s}^{2}=c_{d}^{2}+d_{0}^{2}-2c_{d}d_{0}\;\cdot \;\cos\left(\frac{\pi }{2}-\theta_{2s}\right)\to \;\theta_{2s}=\frac{\pi }{2}-arccos\left(\frac{c_{d}^{2}+d_{0}^{2}-e_{s}^{2}}{2c_{d}d_{0}}\right)$$


-the triangle formed by lines *a*_d_, *e*_s_ and *b*_0_:


(20)
$$e_{s}^{2}=a_{d}^{2}+b_{0}^{2}-2a_{d}b_{0}\;\cdot \;\cos\left(\frac{\pi }{2}-\theta_{3s}\right)\to \;\theta_{3s}=\frac{\pi }{2}-arccos\left(\frac{a_{d}^{2}+b_{0}^{2}-e_{s}^{2}}{2a_{d}b_{0}}\right)$$


-the triangle formed by lines *b*_0_, *f*_s_ and *c*_d_:


(21)
$$f_{s}^{2}=b_{0}^{2}+c_{d}^{2}-2b_{0}c_{d}\;\cdot \;\cos\left(\frac{\pi }{2}+\theta_{4s}\right)\to \;\theta_{4s}=-\frac{\pi }{2}+arccos\left(\frac{b_{0}^{2}+c_{d}^{2}-f_{s}^{2}}{2b_{0}c_{d}}\right)$$


The mean value of the partial strain deformation angle is determined by Formula (22).
(22)$$\Theta_{s}=\frac{1}{n}\overset{n=4}{\underset{i=1}{\sum }}\left|\Theta_{is}\right|$$

Although the partial angle of strain deformation helps determine the shear modulus in further analysis, the total strain angle deformation (consisting of shear and flexural deformation) allows for calculating the stiffness of entire walls.

## 3. Results

### 3.1. Crack Morphology and Digital Image Correlation

After the research, a detailed inventory of the cracks and damages was done—Figure 11.

The propagation of the cracks was captured using the DIC system—Figure 12 and Figure 13. Figure 14 shows the wall displacement with the door opening along the horizontal force (a) and the displacement from the out-of-plane (b).

The cracks on both models were similar. Initial cracks formed at the door opening corners where tensile strength occurred—Figure 11a,c and Figure 12a. The cracks appeared through the bed joints on which the lintel rested and along one head joint. A load increase caused cracks in the wall without openings—Figure 11b,d. The cracks occurred through the bed and head joints and covered about half of the length of the wall—Figure 13a,b.

Diagonal cracks of the vertical pillars appeared during the load increase, and the previously formed damages in the corner developed towards the top ring beam—Figure 11a,c and Figure 12b. The cracks in the vertical pillars occurred from the load side up to approximately 75% of the wall length. After reaching the maximum load, the cracks in the wall without holes (Figure 13c,d) and damage in the pillars between the holes reached almost the entire diagonal length. In addition, cracks formed in the pillars near the base of the wall. An apparent increase in the displacements of the stiffening walls resulted in the cracking of the lower corners of the walls—Figure 11.

### 3.2. Behavior of Stiffening Walls

The behavior of the stiffening walls is shown in the diagrams of the horizontal force- deformation angle relationships—Figure 15 and Figure 16.

In the first MB-AAC-010/1 model, in wall A, full results could not be obtained due to problems with the LVDT sensors. The force causing the crack in the corner of the opening (wall A) was equal *H*_cr,1_ = 13.66 kN. In the wall without holes (wall B), the cracks appeared at force *H*_cr_ = 49.49 kN (first model) and *H*_cr_ = 46.39 kN (second model). The maximum force for the MB-AAC-010/1 model was equal *H*_u_ = 58.34 kN, and for the MB-AAC-010/2 model, the maximum force was equal *H*_u_ = 69.25 kN. In the post-elastic phase, the deformations increase at the stabilized force. The residual forces were determined at approximately70–80% of the maximum forces and were equal *H*_re, model 1_ = 46.96 kN and *H*_re, model 1_ = 49.66 kN (walls B without door opening). The deformation angles in the stiffening walls (A and B) were greater than in the perpendicular walls (1 and 2). The deformation angle at the maximum force was *Θ*_u_ = 0.76 mrad (wall B in the second model); in walls 1 and 2, deformation angles were equal to about *Θ*_u_ = 0.17 and *Θ*_u_ = 0.13 mrad. Detailed test results for the stiffening walls are shown in Table 3. An attempt to capture the behavior phases is shown in Figure 17.

## 4. Analysis of Research Results

### 4.1. Proposition of an Empirical Method of Load Distribution on Stiffening Walls

In line with the purpose of the work, an attempt was made to determine the values of horizontal forces acting on individual stiffening walls (walls A and B located along the load direction). Such forces depend on the stiffness defined as the quotient of the total load corresponding to the displacement—Formula (23):(23)$$K=\frac{H}{∆}=\frac{H}{\theta \;\cdot \;h}\;$$

In which: *H*—total load acting on the wall; ∆—horizontal displacement; θ—shear deformation angle; *h*—wall height.

The total stiffness of the building can be calculated as:(24)$$K_{\mathrm{tot}}=\frac{H_{\mathrm{tot}}}{{∆}_{tot}}=\frac{H_{\mathrm{tot}}}{\theta_{\mathrm{mv}}\;\cdot \;h}\;$$

$H_{\mathrm{tot}}$—total load acting on a building; ${∆}_{tot}$—total displacement of a building; $\theta_{mv}$—mean value of deformation angle of stiffening walls (walls located along horizontal load direction) calculated according to Formula (25); *h*—wall height.
(25)$$\theta_{\mathrm{mv}}=\frac{\theta_{A}+\theta_{B}}{2}\;$$

The total stiffness of the building was calculated in each behavior phase. The results are summarized in Table 4 and Table 5.

In order to determine the load acting on individual stiffening walls, relationships that determine the proportions of the shear deformation angles in *Θ*_A_/*Θ*_B_ = ω as a function of the acting load and shown in Figure 18.

The forces in the stiffening walls balance the total force acting on the building, and the equilibrium condition is as follows:(26)$$H_{\mathrm{tot}}=H_{A}+H_{B}$$

Formula (26) can be developed as:(27)$$K_{\mathrm{tot}}\;\cdot \;{∆}_{\mathrm{tot}}=K_{A}\;\cdot \;{∆}_{A}+K_{B}\;\cdot \;{∆}_{B}\;=K_{\mathrm{tot}}\;\cdot \;\theta_{\mathrm{mv}}=K_{A}\;\cdot \;\theta_{A}+K_{B}\;\cdot \;\theta_{B}=K_{A}\;\cdot \;\theta_{B}\;\cdot \;\omega +K_{B}\;\cdot \;\theta_{B}=\left(K_{A}\;\cdot \;\omega +K_{B}\right)\theta_{B}\to \;\frac{K_{\mathrm{tot}}}{\left(K_{A}\;\cdot \;\omega +K_{B}\right)}=\frac{\theta_{B}}{\theta_{\mathrm{mv}}}$$

Having regard to that:(28)$$K_{B}=\frac{H_{B}}{\theta_{B}\;\cdot \;h}\to \theta_{B}=\frac{H_{B}}{K_{B}\;\cdot \;h}$$

The value of the force acting on the stiffening wall B can be calculated from Formula (29):(29)$$\frac{K_{\mathrm{tot}}\;\cdot \;\theta_{\mathrm{mv}}}{\left(K_{A}\;\cdot \;\omega +K_{B}\right)}=\theta_{B}=\frac{H_{B}}{K_{B}\;\cdot \;h}\;H_{B}=K_{B}\;\cdot \;h\frac{K_{\mathrm{tot}}\;\cdot \;\theta_{\mathrm{mv}}}{\left(K_{A}\;\cdot \;\omega +K_{B}\right)}$$

The value of the force in wall A can be determined in the same way. A summary of the values of the forces in each behavior phase is presented in Table 6.

As the load increases, the degradation of stiffness is significant, as well as the redistribution of internal forces. A wall without an opening takes greater force than a perforated one. After gaining maximum load-bearing capacity, the differences between walls are minor.

### 4.2. Determination of Internal Forces by the Analytical Method

In addition to the empirical method of determining internal forces in stiffening walls, the author’s [6] method was used to calculate the stiffness and load distribution. The shear forces in the walls were determined from the following formulas:shear forces due to the load *H*_x_ and *H*_y:_
(30)$$H_{x,i}=H_{x}\frac{K_{y,i}}{\sum_{i}K_{y,i}},\;\;H_{y,i}=H_{y}\frac{K_{x,i}}{\sum_{i}K_{x,i}}\;\;$$

shear forces induced by torsional moments M_sx_ and M_sy_:


(31)
$$H_{\mathrm{xs},i}=\pm M_{\mathrm{sx}}\frac{{\bar{a}}_{\mathrm{xi}}K_{y,i}}{\sum_{i}{\bar{a}}_{\mathrm{xi}}^{2}K_{x,i}+\sum_{i}{\bar{a}}_{\mathrm{yi}}^{2}K_{y,i}},\;\;H_{\mathrm{ys},i}=\pm M_{\mathrm{sx}}\frac{{\bar{a}}_{\mathrm{xi}}K_{y,i}}{\sum_{i}{\bar{a}}_{\mathrm{xi}}^{2}K_{x,i}+\sum_{i}{\bar{a}}_{\mathrm{yi}}^{2}K_{y,i}},\;\;H_{\mathrm{xs},i}=\pm M_{\mathrm{sy}}\frac{{\bar{a}}_{\mathrm{xi}}K_{y,i}}{\sum_{i}{\bar{a}}_{\mathrm{xi}}^{2}K_{x,i}+\sum_{i}{\bar{a}}_{\mathrm{yi}}^{2}K_{y,i}},\;\;H_{\mathrm{ys},i}=\pm M_{\mathrm{sy}}\frac{{\bar{a}}_{\mathrm{xi}}K_{y,i}}{\sum_{i}{\bar{a}}_{\mathrm{xi}}^{2}K_{x,i}+\sum_{i}{\bar{a}}_{\mathrm{yi}}^{2}K_{y,i}},$$


In which: ${\bar{a}}_{\mathrm{xi}},{\bar{a}}_{\mathrm{yi}}$—distances between the gravity center of the wall fragments and the rotation centre (RC); *h*_m_—height of the wall.

The bending moments are:bending moments due to load *H*_x_ and *H*_y_:
(32)$$M_{\mathrm{ox},i}=M_{\mathrm{ox}}\frac{K_{x,i}}{\sum_{i}K_{x,i}},\;\;M_{\mathrm{oy},i}=M_{\mathrm{ox}}\frac{K_{y,i}}{\sum_{i}K_{y,i}},\;\;$$

bending moments due to torsional moments of the building *M*_sx_ and *M*_sy_:


(33)
$$M_{\mathrm{sx},i}=\pm H_{\mathrm{xs},i}h_{m},\;\;M_{\mathrm{sy},i}=\pm H_{\mathrm{ys},i}h_{m}\;$$


The coordinates of the rotation centre (RC) are derived from the equilibrium of forces according to the following Formula (34):(34)$$x_{R}=\frac{\sum_{i}(a_{\mathrm{xi}}K_{\mathrm{xi}})\;}{\sum_{i}K_{\mathrm{xi}}\;},\;\;y_{R}=\frac{\sum_{i}(a_{\mathrm{yi}}K_{\mathrm{yi}})\;}{\sum_{i}K_{\mathrm{yi}}\;},$$

In which: *a*_xi_, *a*_yi_—the distance between the load centre (LC) and the rotation center of the wall or stiffening group, *K*_xi_, *K*_yi_—stiffness of the wall or stiffening group.

After calculating the stiffness of the wall with an opening—a single wall was divided into zones weakened by perforation and zones of greater stiffness (lintel and bottom spandrel)—Figure 19. The displacement from the unit load of the upper edge of the wall with concentrated force and the bending moment was determined, analogously to Canadian regulations [62]. The total displacement of the upper edge of the wall is the sum of the displacements of the bottom spandrels, inter-opening pillars and lintels—Formula (35):∆_w_ = ^A^∆_w_ + ^P^∆_w_ + ^B^∆_w_(35)

In which: ^A^∆_w_, ^B^∆_w_—displacement of the bottom spandrel and lintel_,_ ^P^∆_w_—displacement of the wall with the opening of height *h*_0_ and length *l*.

The displacements of the wall components depend on the geometry and boundary conditions. If the height ratio to the wall’s length is *h*/*l* > 2, the effects of tangential stresses in determining the wall stiffness can be neglected. Otherwise, the stiffness should be calculated, considering shear deformations. Wall stiffness depending on the boundary conditions is shown in Table 7.

After calculating the total displacement of the wall, its stiffness can be estimated following the Formula (36):(36)$$K_{w}=\frac{1}{{∆}_{w}}$$

∆*w*—is the total displacement of the top edge of the wall due to unit load *H* = 1.

The analytical calculations of the stiffening walls were carried out according to the following procedure:The length of the transverse wall fragment *b*_eff1_ was assumed following the recommendations of Eurocode 6 [1].Wall with the opening was divided into fragments as shown in Figure 19. Moments of inertia of the wall components were calculated, taking into account the transverse *b*_eff1_ parts.Static schemes of each component were established: “C”—cantilever wall, “F”—restrained wall.The stiffness *K* of the wall components were determined according to the formulas in Table 7.The stiffness of walls was determined according to Figure 19 and dependence (35),The distances *a*_xi_, *a*_yi_ to the load center (LC) were assumed.The localization of the rotation center was calculated according to Formula (34).The internal forces in walls were calculated according to Formulas (30)–(33).

The results of the geometric data and wall stiffness calculations are given in Table 8. Based on Formula (34), the coordinates of the torsion center are: *x*_R_ = 0 m, *y*_R_ = −0.61 m. Based on Formulas (30)–(31), the internal forces ^cal^*H*_cr_ and ^cal^*H*_u_ in walls were calculated—Table 9.

The best convergence of the empirical load distribution on the walls with the analytical model was obtained at the crack phase. The most significant discrepancies in results were obtained at the maximum load in the wall. The estimated shear force in the linear-elastic model was over 24% lower than the empirical value (test) in the wall with a window opening. The opposite situation occurred in the wall without the door opening, and the empirical load estimation was almost 16% lower. There was internal forces redistribution from a wall without an opening (stiffer) to a wall of lower stiffness with an opening. This phenomenon occurs when the walls differ in their initial stiffness.

## 5. Discussion and Conclusions

The article presents the results of tests of two models of buildings made of masonry units made of AAC. Stiffening walls with and without a door opening were erected along the load direction. The studies analyzed the values of the forces acting on the building and the shear deformation angles. Based on the damage morphology, it was shown that the first crack formed in the tensile corner of the door opening, and the subsequent cracks formed in the wall without a hole. A load increase caused the crack propagation. The values of the total stiffness of the building and the stiffness of individual walls were determined empirically based on elementary relationships. Based on the changes in the value of the shear deformation angles, the phases of work of the stiffening walls were determined:the initial phase (cracks in the tensile corner); 0—*H*_cr,1_,elastic phase (cracks in the wall without opening); *H*_cr,1_—*H*_cr_,nonlinear phase (up to maximum horizontal force); *H*_cr_—*H*_u_,post-peak residual phase (decrease the horizontal force and stabilization of shear deformations); *H*_u_—*H*_re_.

Two approaches to the determination of forces in stiffening walls have been proposed:(a)empirical approach—based on empirical proportions between the deformation an-gles of the walls,(b)analytical approach—based on determining the stiffness of each wall based on its components [6].

The results were compared with the empirical ones. The most remarkable differences were found at the maximum load level. The force values decreased in the wall with initially higher stiffness, and in the wall with a door opening, they increased. The obtained result proves the potential redistribution of internal forces of 25%, which is over 10% more than the recommendations of Eurocode 6. Further work in this area will concern the determination of the level of loads at which the permissible redistribution of loads of 15% is achieved.

The presented research results are only a part of a more extensive study of stiffening walls in masonry buildings, including the new concept of analytical models and extensive Finite Element Method analysis. The research will define and verify the algorithm for determining the redistribution of internal forces in the stiffening walls.

## Figures and Tables

**Figure 1 materials-15-07404-f001:**
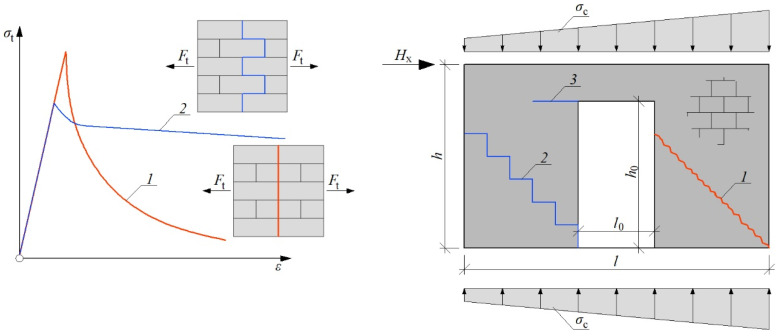
Behavior of stiffening masonry wall in simple and complex failure mode; *1*—straight crack (high compressive stress); *2*—stepped crack (low compressive stress); *3*—tensile corner; *s_c_*—compressive stress; *s_t_*—tensile stress; *H*_x_—horizontal shear force; *F*_t_—tensile force.

**Figure 2 materials-15-07404-f002:**
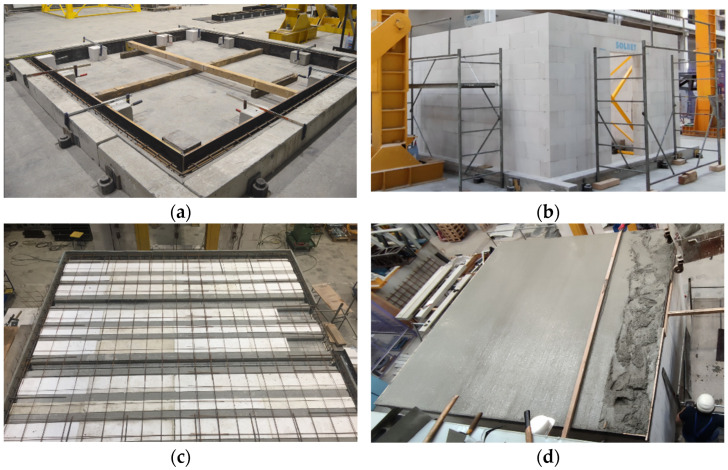
Construction stages; (**a**) formwork of the lower perimeter beam; (**b**) erection of masonry walls made of autoclaved aerated concrete; (**c**) slab top reinforcement; (**d**) concreting the slab.

**Figure 3 materials-15-07404-f003:**
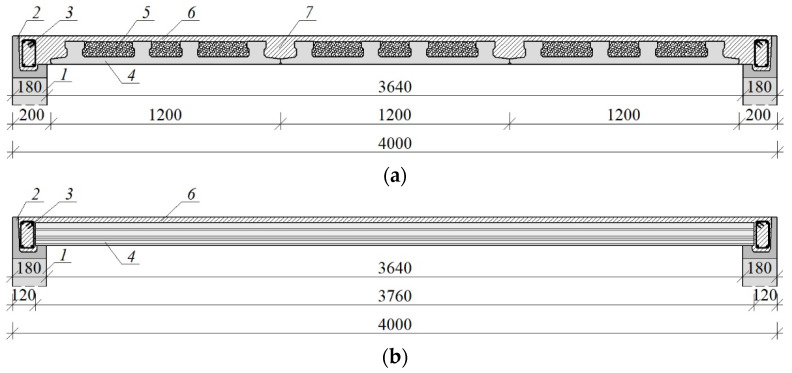
Slab construction; (**a**) cross-section of panels; (**b**) longitudinal section of the panels; *1*—masonry wall made of autoclaved aerated concrete; *2*—precast L-shaped element made of lightweight concrete; *3*—reinforcement of the circumferential top beam; *4*—precast, pretensioned panel slab; *5*—lightweight filling–polystyrene; *6*—concrete overlay; *7*—the monolithic connection between panels.

**Figure 4 materials-15-07404-f004:**
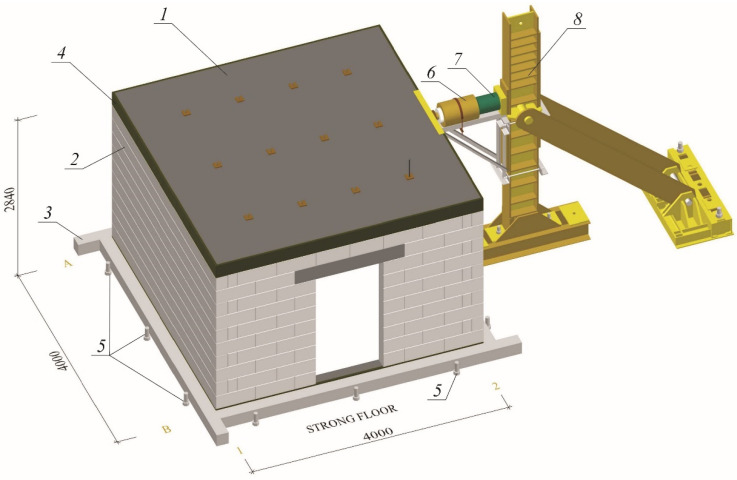
Research model in the test stand; *1*—slab; *2*—masonry wall made of autoclaved aerated concrete units; *3*—ring bottom beam; *4*—ring top beam; *5*—fixing the building model in the slab of great forces; *6*—hydraulic cylinder; *7*—force gauge; *8*—steel column.

**Figure 5 materials-15-07404-f005:**
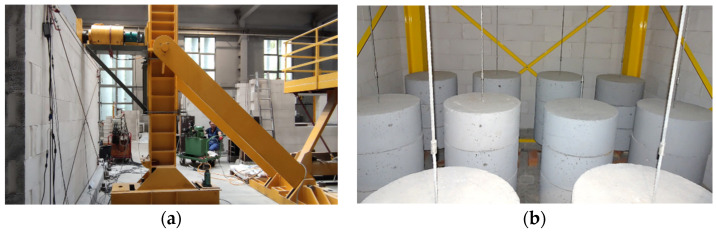
Loads of single-storey buildings; (**a**) horizontal load; (**b**) vertical load.

**Figure 6 materials-15-07404-f006:**
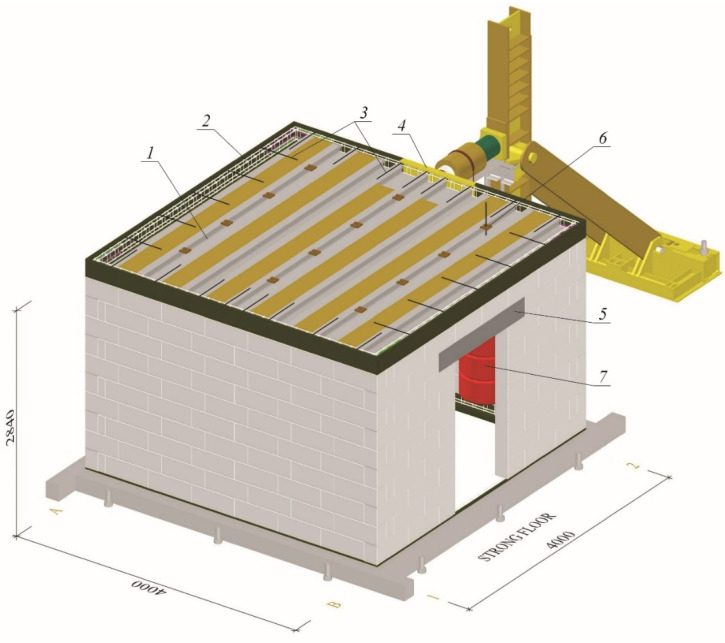
Research model in the test stand; *1*—prestressed precast slab panel; *2*—reinforcement of the top ring beam; *3*—top reinforcement of the slab; *4*—steel C-profile; *5*—precast lintel; *6*—load suspension points; *7*—visible weights.

**Figure 7 materials-15-07404-f007:**
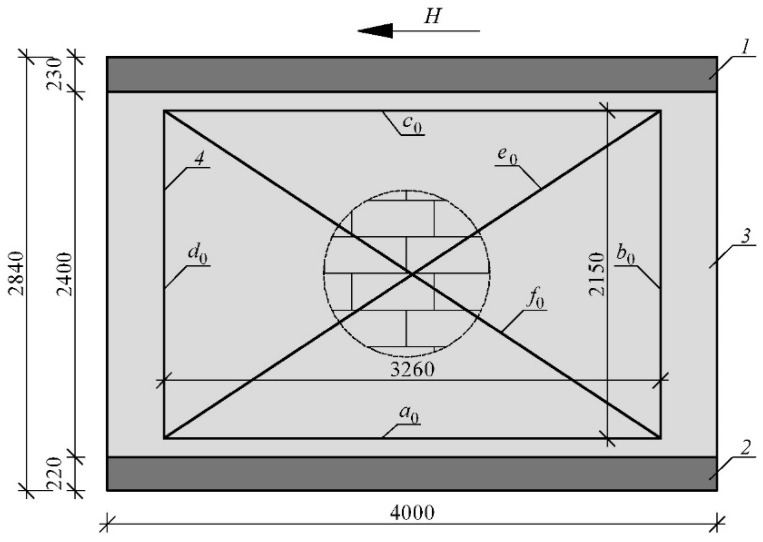
Frame system mounted on each model’s walls to measure shear strain and deformation angle; *H*—horizontal shear force; *1*—rigid diaphragm; *2*—concrete foundation; *3*—masonry wall made of AAC; *4*—measuring base; *a*_0_, *c*_0_—horizontal part of the frame system; *b*_0_, *d*_0_—vertical part of the frame system; *e*_0_, *f*_0_—diagonal part of the frame system.

**Figure 8 materials-15-07404-f008:**
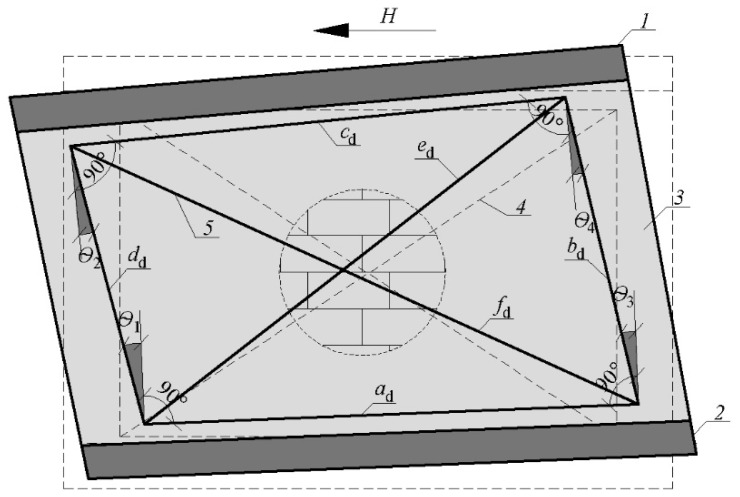
Deformation of measuring base (frame system) due to the action of shear force; *H*—horizontal shear force; *1*—rigid diaphragm; *2*—concrete foundation; *3*—masonry wall made of AAC; *4*—measuring base before deformation; *5*—measuring base after deformation; *a*_d_, *c*_d_—deformed horizontal part of the frame system; *b*_d_, *d*_d_—deformed vertical part of the frame system; *e*_d_, *f*_d_—deformed diagonal part of the frame system.

**Figure 9 materials-15-07404-f009:**
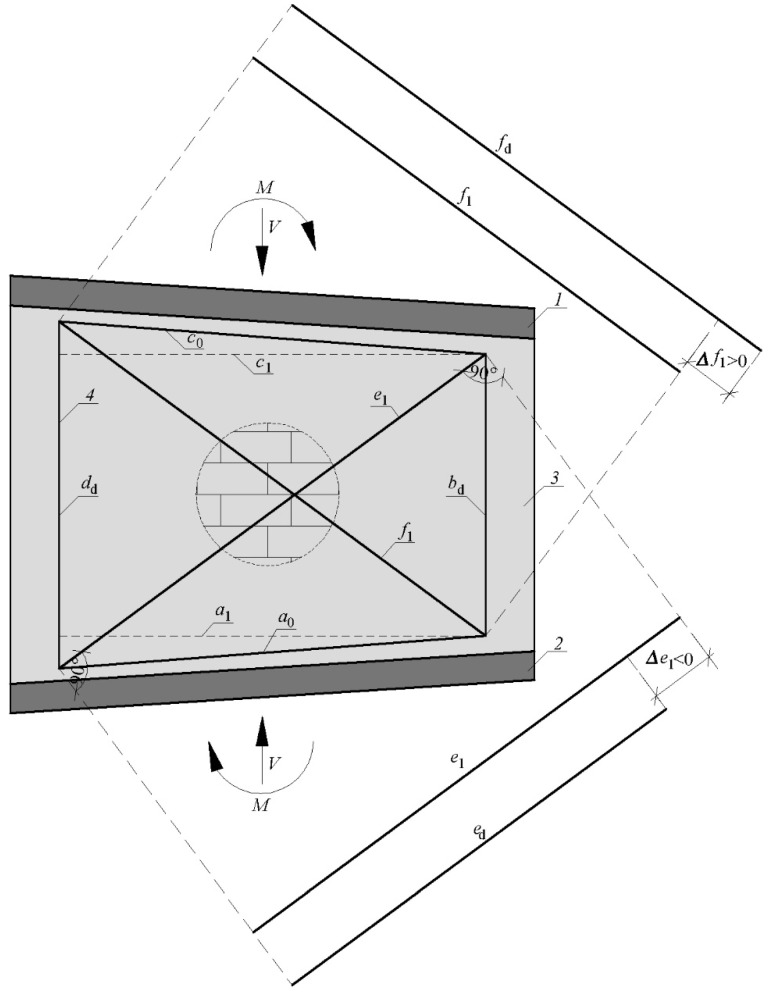
Deformation of measuring base resulting from flexural deformation; *V*—vertical force; *M*—in-plane bending moment; *1*—rigid diaphragm; *2*—concrete foundation; *3*—masonry wall made of AAC; *4*—measuring base before deformation; *a*_0_, *c*_0_—undeformed horizontal part of the frame system; *e*_1_, *f*_1_—deformed diagonal part of the frame system (result from flexural deformation).

**Figure 10 materials-15-07404-f010:**
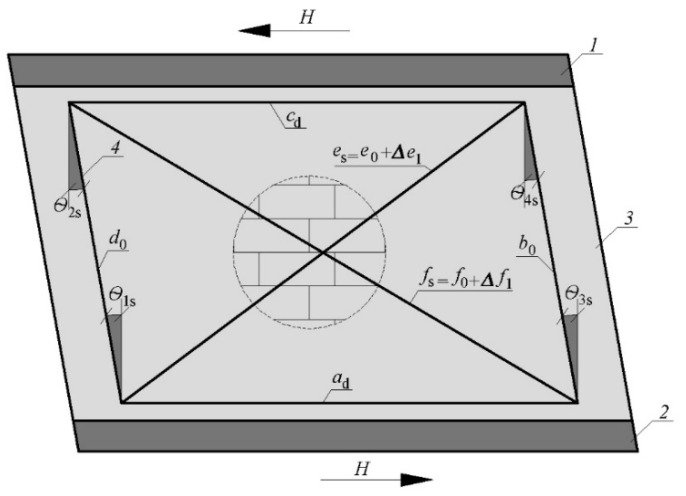
Deformation of measuring base resulting from shear deformation; *H*—horizontal shear force; *1*—rigid diaphragm; *2*—concrete foundation; *3*—masonry wall made of AAC; *4*—measuring base before deformation, *5*—measuring base after deformation; *a*_d_, *c*_d_—deformed horizontal part of the frame system; *b*_0_, *d*_0_—undeformed vertical part of the frame system; *e*_s_, *f*_s_—deformed diagonal part of the frame system (result from shear deformation); $\theta_{1s},\theta_{2s},\theta_{3s},\theta_{4s}$—values of partial strain deformation angles.

**Figure 11 materials-15-07404-f011:**
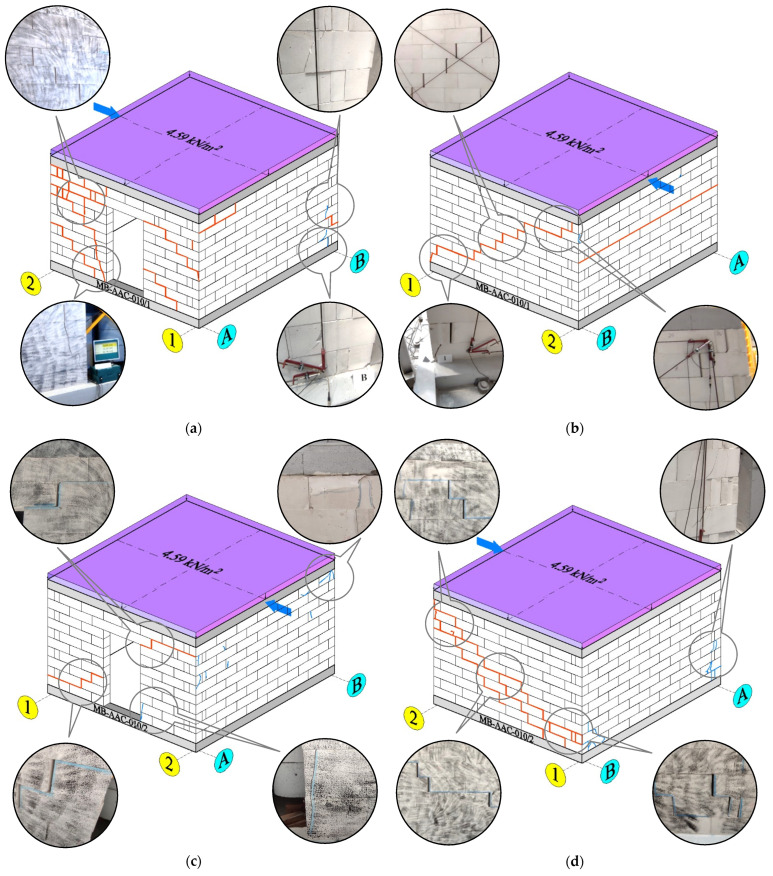
The crack pattern of tested masonry buildings; (**a**) front view of MW-AAC-010/1 model; (**b**) back view of MW-AAC-010/1 model; (**c**) front view of MW-AAC-010/2 model; (**d**) back view of MW-AAC-010/2 model; the blue arrow marks the horizontal load.

**Figure 12 materials-15-07404-f012:**
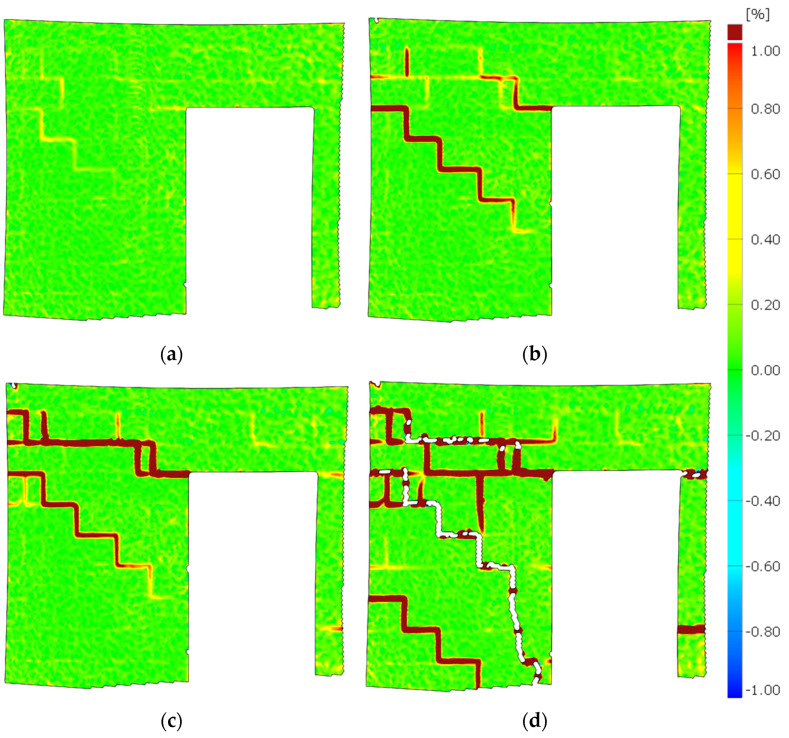
The crack propagation—MW-AAC-010/1 model; (**a**) crack pattern—*H*_x_ = 32.23 kN; (**b**) crack pattern—*H*_x_ = 46.39 kN; (**c**) crack pattern—*H*_x_ = 58.22 kN; (**d**) crack pattern—*H*_x_ = 42.40 kN (post-peak behavior).

**Figure 13 materials-15-07404-f013:**
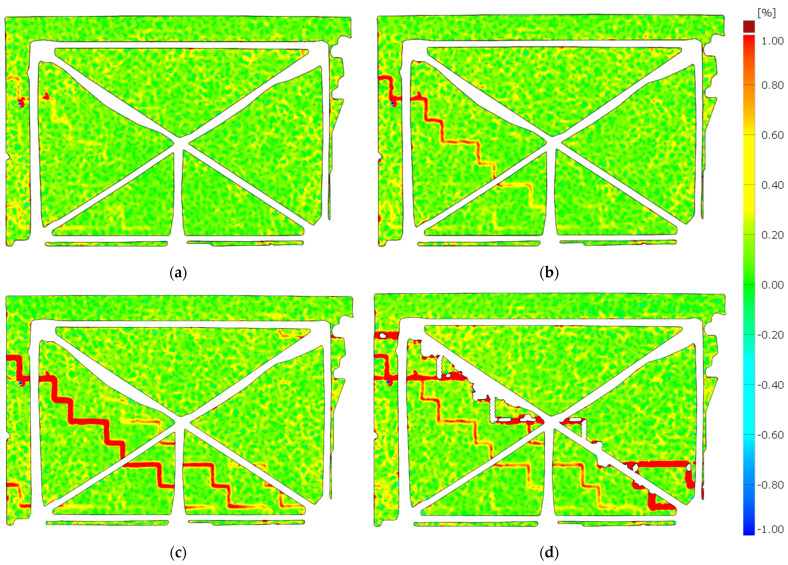
The crack propagation—MW-AAC-010/2 model; (**a**) crack pattern—*H*_x_ = 48.05 kN; (**b**) crack pattern—*H*_x_ = 48.60 kN; (**c**) crack pattern—*H*_x_ = 69.25 kN; (**d**) crack pattern—*H*_x_ = 42.77 kN (post-peak behavior).

**Figure 14 materials-15-07404-f014:**
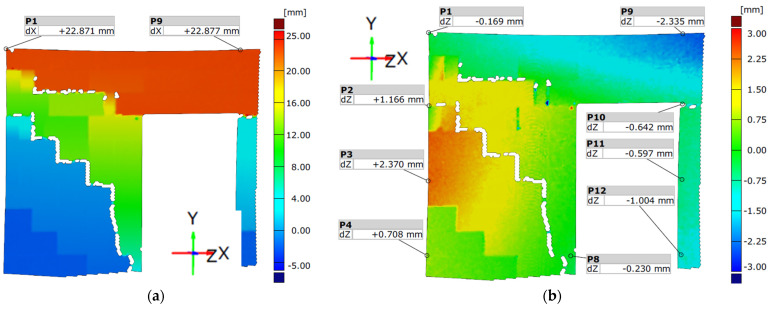
Post peak deformation analysis—MW-AAC-010/1 model—*H*_x_ = 42.40 kN; (**a**) displacement along the *X* axis; (**b**) displacement along the *Z* axis.

**Figure 15 materials-15-07404-f015:**
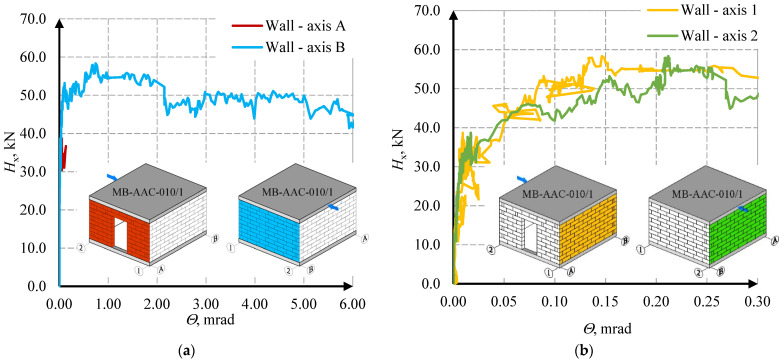
Relationship between horizontal force and strain deformation angle for MW-AAC-010/1 model; (**a**) results for A and B wall (**b**) results for 1 and 2 wall.

**Figure 16 materials-15-07404-f016:**
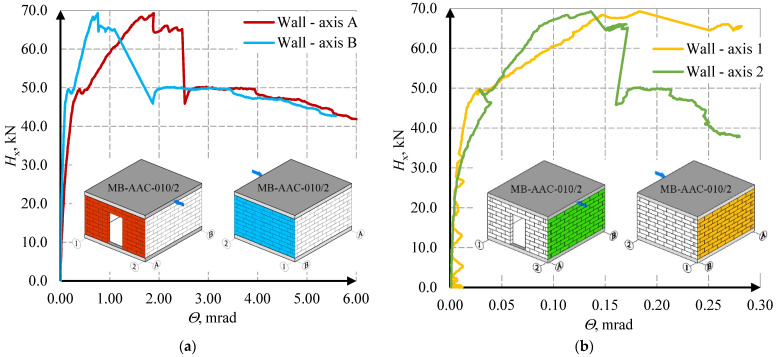
Relationship between horizontal force and strain deformation angle for MW-AAC-010/2 model; (**a**) results for A and B wall (**b**) results for 1 and 2 wall.

**Figure 17 materials-15-07404-f017:**
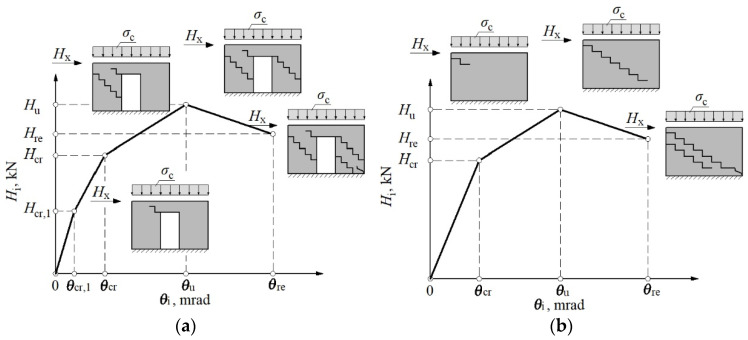
Behavior phases of stiffening walls; (**a**) wall with door opening (**b**) wall without door opening.

**Figure 18 materials-15-07404-f018:**
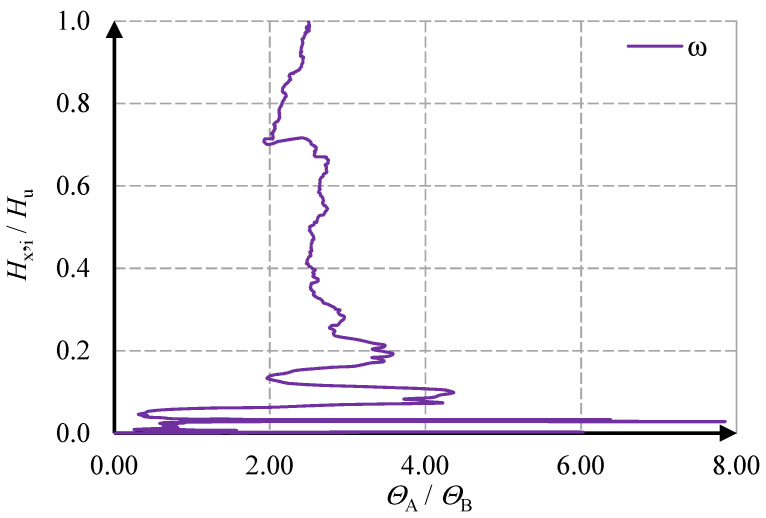
Relation between the *H*_x,i_/*H*_u_ and *Θ*_A_/*Θ*_B_ for the MB-AAC-010/2 model.

**Figure 19 materials-15-07404-f019:**
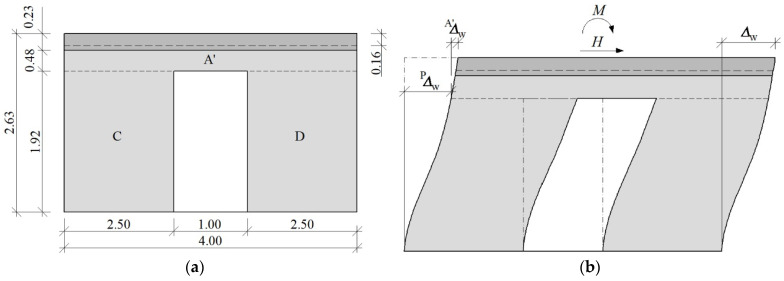
Total wall stiffness method [6]: (**a**) division of a wall with openings into component elements, (**b**) wall deformation caused by horizontal loads.

**Table 1 materials-15-07404-t001:** Vertical loads of research models—details.

Type of Load	Description	Value, kN
dead	self-weight of slab	57.00
live	one weight	2.04
	load per one steel rod	6.12
	total load of 12 steel rods	73.44

**Table 2 materials-15-07404-t002:** Initial compressive stresses in the masonry walls.

Description	Value
total vertical load on the walls, *P*_c_	130.44 kN
the area of the horizontal layout of the walls, *A*_c_	2.82 m^2^
compressive prestress of the walls, *σ*_c_	46,33 kPa (~0.05 MPa)

**Table 3 materials-15-07404-t003:** Summary of test results.

Model	Model Wall	Initial Phase		Elastic Phase		Nonlinear Phase		Post-Peak Residual Phase	
		*H*_cr,1_, kN	*Θ*_cr,1_, mrad	*H*_cr_, kN	*Θ*_cr_, mrad	*H*_u_, kN	*Θ*_u_, mrad	*H_r_*_e_, kN	*Θ*_re_, mrad
MB-AAC-010/1	with door opening (wall A)	13.66	0.001	-	-	58.34	-	-	-
	without door opening (wall B)	-	-	49.49	0.07		0.74	46.96	2.35
MB-AAC-010/2	with door opening (wall A)	9.76	0.02	49.51	0.38	69.25	1.89	50.04	2.84
	without door opening (wall B)	-	-	46.39	0.10		0.76	49.66	2.03

**Table 4 materials-15-07404-t004:** Stiffness of the building model MB-AAC-010/2.

Model	Model Wall	Initial Phase		Elastic Phase		Nonlinear Phase		Post-Peak Residual Phase	
		*Θ*_cr,1,mv_, mrad	*K*_tot,cr,1_, kN/mm	*Θ*_cr,mv_, mrad	*K*_tot,cr_, kN/mm	*Θ*_umv_, mrad	*K*_tot,u_, kN/mm	*Θ*_res,mv_, mrad	*K*_tot,re_, kN/mm
MB-AAC-010/2	Wall A and B	-	-	0.24	76.01	1.32	19.90	2.43	7.79

**Table 5 materials-15-07404-t005:** Stiffness of stiffening walls based on the test.

	Model Wall	Initial Phase	Elastic Phase	Nonlinear Phase	Post-Peak Residual Phase
		*K*_tot,cr,1_, kN/mm	*K*_tot,cr_, kN/mm	*K*_tot,u_, kN/mm	*K*_tot,re_, kN/mm
MB-AAC-010/1	with door opening (wall A)	5979.31	-	-	-
	without door opening (wall B)	-	255.15	29.84	7.58
MB-AAC-010/2	with door opening (wall A)	187.60	49.33	13.93	6.70
	without door opening (wall B)	-	179.86	34.82	9.32

**Table 6 materials-15-07404-t006:** Forces acting on stiffening walls in the MB-AAC-010/2 model.

	Model Wall	Initial Phase	Elastic Phase	Nonlinear Phase	Post-Peak Residual Phase
		*H*_cr,1_, kN	*H*_cr_, kN	*H*_u_, kN	*H*_re_, kN
MB-AAC-010/2	with door opening (wall A)	-	19.27	34.62	17.05
	without door opening (wall B)	-	28.67	34.62	32.80
	*H_A_/H_B_*	-	0.67	1.00	0.52

**Table 7 materials-15-07404-t007:** Stiffness of walls subjected to shear with bending.

Static Scheme	*h/l* ≤ 2		*h/l* ≥ 2	
	Force *P*	Moment *M*	Force *P*	Moment *M*
Cantilever type “C”	$K_{p}=\frac{1}{\frac{h_{m}^{3}}{3EI}+\frac{1.2h_{m}}{GA}}$	$\frac{2EI}{h_{m}^{2}}$	$K_{M}=\frac{3EI}{h_{m}^{3}}$	$K_{M}=\frac{2EI}{h_{m}^{2}}$
	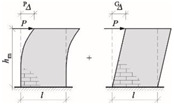	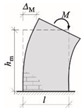	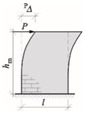	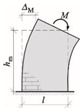
Double- fixed type “F”	$K_{p}=\frac{1}{\frac{h_{m}^{3}}{12EI}+\frac{1.2h_{m}}{GA}}$	--	$K_{M}=\frac{12EI}{h_{m}^{3}}$	--
	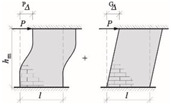	--	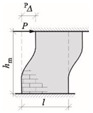	

**Table 8 materials-15-07404-t008:** Geometric and stiffness characteristics of walls.

Wall or Component		Moment of Inertia *I*, m^4^	Surface *A*, m^2^	Static Scheme	Distance from the Center of Gravity of the Wall to a Point LC *a*, m	Stiffness *K*, MN/m	
1		*I*_x2_ = 1.59	4.18	“F”	*a*_x2_ = 1.91		
2		*I*_x2_ = 1.59	4.18	“F”	*a*_x2_ = 1.91	*K*_x,2_ = 114.0	
A	A’	*I*_yA’_ = 1.59	4.18	“F”	*a*_xA_ =1.91	*K*_y,A’_ = 592.8	*K*_y,A_ = 81.5
	C	*I*_yC_ = 0.09	0.09	“F”		*K*_y,C_ = 47.2	
	D	*I*_yD_ = 0.09	0.09	“F”		*K*_y,D_ = 47.2	
B		*I*_yB_ = 1.5	4.18	“F”	*a*_xB_ = –1.91	*K*_y,B_ = 114.0	

**Table 9 materials-15-07404-t009:** Comparison of test results and analytical analysis.

	Model Wall	Elastic Phase			Nonlinear Phase		
		*H*_cr_, kN	^cal^*H*_cr_, kN	*H*_cr_/^cal^*H*_cr_	*H*_u_, kN	^cal^*H*_u_, kN	*H*_u_/^cal^*H*_u_
MB-AAC-010/2	with door opening (wall A)	19.27	19.38	0.99	34.62	27.99	1.24
	without door opening (wall B)	28.67	28.56	1.00	34.62	41.25	0.84
	sum	47.94	47.94	-	69.24	69.24	-

## Data Availability

Not applicable.

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
