# Peer review of "Research on the Behavior of Stiffening Walls in Single-Storey Buildings Made of Autoclaved Aerated Concrete (AAC) Masonry Units"

_materials, 2022, doi:10.3390/ma15207404_

Round 1

Reviewer 1 Report

The article presents the test results of two building models made of AAC masonry units. The study analyzes the force values and shear deformation angles acting on the building. And a method of load redistribution is proposed. The structure of the article is complete and the content is detailed and innovative, but there are still some small problems in the article:

1. The layout of tables and pictures is too messy and not beautiful

2. The introduction cites references from the last year or two;

3. The conclusion is simplified and some key points are highlighted

In summary, the article is received after further revision

Author Response

Review 1

Dear Reviewer,

We would like to thank you for taking the time to review the article and comments. As suggested, the article was supplemented and corrected according to the recommendations of the reviews.

  1. We tried to improve the layout of tables and figures in a more readable way. We highlighted all changes in yellow.
  2. The subject of stiffening walls is a poorly recognized issue in the literature. Other literature in this field known to us is provided below:
  • Meftah, A.; Tounsi, A.; El Abbas, A.B. A Simplified Approach for Seismic Calculation of a Tall Building Braced by Shear Walls and Thin-Walled Open Section Structures. Eng. Struct. 2007, 29, 2576–2585, doi:10.1016/j.engstruct.2006.12.014.
  • Aksogan, O.; Bikce, M.; Emsen, E.; Arslan, H.M. A Simplified Dynamic Analysis of Multi-Bay Stiffened Coupled Shear Walls. Adv. Softw. 2007, 38, 552–560, doi:10.1016/j.advengsoft.2006.08.019.
  • Wdowicki, J.; Wdowicka, E. Three-Dimensional Analysis of Asymmetric Shear Wall Structures with Connecting and Stiffening Beams. Struct. 2012, 42, 362–370, doi:10.1016/j.engstruct.2012.04.038.
  • Sreejith, P.P.; Sivan, P.P.; Praveen, A.; Gajendran, C.; Nisha, V. Simplified Method for Shear Strength Prediction of Confined Masonry Walls Subjected to in Plane Loads. Procedia Technol. 2016, 24, 155–160, doi:10.1016/j.protcy.2016.05.022.
  • Minaie, E.; Moon, F.L.; Hamid, A.A. Nonlinear Finite Element Modeling of Reinforced Masonry Shear Walls for Bidirectional Loading Response. Finite Elem. Anal. Des. 2014, 84, 44–53, doi:10.1016/j.finel.2014.02.001.
  • Nollet and Smith, B.S. Stiffened-Story Structure Tall. Comput. Struct. 1998, 66, 225–240.
  • Shi, Q.-X.; Wang, B. Simplified Calculation of Effective Flange Width for Shear Walls with Flange. Struct. Des. Tall Spec. Build. 2016, 25, 558–577, doi:10.1002/tal.1272.
  • Sajid, H.U.; Ashraf, M.; Ali, Q.; Sajid, S.H. Effects of Vertical Stresses and Flanges on Seismic Behavior of Unreinforced Brick Masonry. Struct. 2018, 155, 394–409, doi:10.1016/j.engstruct.2017.11.013.
  • Tripathy, D.; Singhal, V. Estimation of In-Plane Shear Capacity of Confined Masonry Walls with and without Openings Using Strut-and-Tie Analysis. Struct. 2019, 188, 290–304, doi:10.1016/j.engstruct.2019.03.002.

The remaining publications come from later years and concern current trends in analyses and calculations.

  1. Conclusions were corrected and simplified.
  2. The layout of figures and tables will be further improved at the proofreading stage.

Reviewer 2 Report

Report on the manuscript materials-1936926-peer-review-v1 entitled “Research on the behavior of stiffening walls in single-storey buildings made of autoclaved aerated concrete masonry units”.

The submitted manuscript should be revised. The following points should be addressed

1.  The language of the manuscript should be revised.

2. The crack morphology in figure 11 is not clear.

3. The conclusion part should be rewritten to be more concise and easier understood.

4. High self-citations of the second author.

Author Response

Review 2

Dear Reviewer,

We would like to thank you for taking the time to review the article and comments. As suggested, the article was supplemented and corrected according to the recommendations of the reviews.

  1. All linguistic corrections of the publication are marked in
  2. Changes resulting from the review are marked in purple; remarks of other reviewers in yellow and
  3. The pattern of cracks on all walls of the buildings is sketched (figure 11) after the test. The figure presents cracks that concern the moment of destruction, not the morphology (development of scratches). The propagation were was captured in figure 12 and 13.
  4. Conclusions were corrected and simplified.
  5. Some quotations from the publication of the second author have been removed:
  • Radosław Jasiński; Łukasz Drobiec Comparison Research of Bed Joints Construction and Bed Joints Reinforcement on Shear Parameters of AAC Masonry Walls. Civ. Eng. Archit. 2016, 10, 1329–1343, doi:10.17265/1934-7359/2016.12.00
  • Drobiec, Ł.; Jasiński, R. Influence of the Kind of Mortar on Mechanical Parameters of AAC Masonry Subjected to Shear – the Basic Strength Parameters. (in Polish). Bud. 2015, 5, 106–109, doi:10.15199/33.2015.05.44.
  • Drobiec, Ł.; Jasiński, R. Influence of the Kind of Mortar on Mechanical Parameters of AAC Masonry Subjected to Shear – Dilatational Deformability. (in Polish). Bud. 2015, 7, 116–119, doi:10.15199/33.2015.07.32.

Reviewer 3 Report

The subject of the work presented is interesting and current.

This paper presents the results of two full-scale buildings made of autoclaved aerated concrete (AAC) masonry elements. The presented research results are only a part of  an extensive study of stiffening walls in masonry buildings conducted at the Silesian University of  Technology.

In the attached file are indicated the changes to be made in the article. English must be revised. Some sentences are meaningless.

Author Response

Review 3

Dear Reviewer,

We would like to thank you for taking the time to review the article and comments. As suggested, the article was supplemented and corrected according to the recommendations of the reviews.

  1. All linguistic corrections of the publication are marked in
  2. All changes have been made to the attached file except for the note regarding the description of walls 1 and 2. The tests presented are part of an extensive test program, and the components' designation has already been determined. Stiffening walls are marked with letter symbols; the transverse walls are marked with Arabic numerals.
  3. Changes of other reviewers were marked s in yellow and purple.
